# FastFace: Training-Free Identity Preservation Tuning in Distilled Diffusion via Guidance and Attention

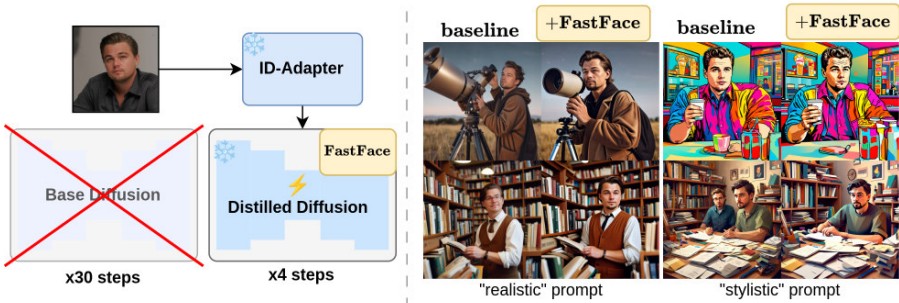

Figure 1: FastFace method framework: on the left - high-level idea of pipeline, enabling few-sep id-preserving generation, on the right - effect of FastFace components on realistic and stylistic generations

## Abstract

The recent proliferation of identity-preserving (ID) adapters has significantly advanced personalized generation with diffusion models. However, these adapters are predominantly co-trained with base diffusion models, inheriting their critical drawback: slow, multi-step inference. This work addresses the challenge of adapting pre-trained ID adapters to much faster distilled diffusion models without requiring any further training. We introduce FastFace, a universal framework that achieves this via two key mechanisms: (1) the decomposition and adaptation of classifier-free guidance for few-step stylistic generation, and (2) attention manipulation within decoupled blocks to enhance identity similarity and fidelity. We demonstrate that FastFace generalizes effectively across various distilled models and maintains full compatibility with a wide range of existing ID-preserving methods, enabling high-fidelity personalized image generation at unprecedented speeds.

## 1 Introduction

Diffusion models have emerged as a dominant paradigm in generative modeling, achieving state-of-the-art performance in high-fidelity image synthesis, with plethora of models coming out in recent years (Ho et al. (2020), Dhariwal & Nichol (2021), Rombach et al. (2022), Podell et al. (2023), Esser et al. (2024), Labs (2024)). Diffusion distillation aims to to speed up inference by reducing number of sampling steps, with a lot of approaches and versions releasing in past several years, such as LCM, Turbo, Lightning, Hyper, and others (Luo et al. (2023), Sauer et al. (2024a), Lin et al. (2024c), Ren et al. (2024), Sauer et al. (2024b)); common results of these distillation are 1) architecture of diffusion model remains the same 2) inference becomes significantly more efficient in terms of number of steps. In parallel, diffusion models have been adapted for task of id-preserving generation, where image with face of a person $c_{id}$ is used as condition, and diffusion can generate images with novel identities without further finetuning (Ye et al. (2023), Li et al. (2024), Wang et al. (2024a), Guo et al. (2024), Jiang et al. (2025)). These methods are commonly trained with base diffusion models and can be denoted as ID-adapters.

Integrating image conditioning with distilled diffusion has recently emerged as a separate problem. Several works (Zhang et al. (2023), Xiao et al. (2023), Parmar et al. (2024)) propose to adapt prior of ControlNet towards new trajectories of distilled models through finetuning. While promising, these works do not propose a universal approach to any model, i.e. for a new model, a completely new algorithmic design is required. In recent work Sun et al. (2024) authors introduce training-free adaptation of image conditioning through classifier guidance, but their approach suffers from requirement for backpropagation during inference.

In contrast to previous work, we aim to develop general, light, training-free mechanisms that can be used in plug-and-play manner to improve quality of id-preserving generation with any distilled diffusion model. We separate identity-preserving generation scenarios and develop two contributions - for stylistic generation we adopt and tune decoupled classifier-free guidance, where conditional noise prediction is splitted into two parallel terms, and for realistic id preservation we introduce attention manipulation in decoupled blocks is transformed via simple analytical functions to be more focused on facial regions during generation. We denote FastFace as joint application of these mechanisms, and it's effect on generated image is visualized in Figure 7 - it achieves superior identity preservation and image fidelity while not losing prompt following.

To demonstrate the effectiveness and generality of FastFace, we conduct a comprehensive evaluation across a range of SDXL-distilled checkpoints and identity adapters. Our empirical results, which demonstrate robust generalization, are enabled by two technical contributions:

Decoupled Classifier-Free Guidance Mechanism: We introduce a guidance strategy that decomposes the network output into semantically interpretable components. This decomposition is specifically tuned to enhance performance in the few-step sampling regime characteristic of distilled models.

Attention Manipulation for Identity Enhancement: We develop an inference-time method to precisely manipulate attention maps within decoupled attention blocks. By strategically reinforcing attention over facial regions, this approach substantially improves identity similarity without any additional training.

We evaluate our method on subset of high-quality OmniContext dataset (Wu et al. (2025), as well as collected and released DiverseFaces benchmark, which includes a wide variety of nationalities, age groups, and genders and decomposes scenarios of stylistic and realistic generation, allowing to use both for specific ablations and general evaluation. The results confirm the superior performance and broad applicability of FastFace for efficient identity-preserving generation.

## 2 RELATED WORK

**ID-preserving generation methods** Identity-preserving generation, as we describe it, is a problem of preserving identity similarity in generation output given an image with the face. A lot of methods came out around this problem, including IpAdapter-FaceID Ye et al. (2023), Photomaker Li et al. (2024), PuLID Guo et al. (2024), InstantID Wang et al. (2024a). They differ in their overall approaches and flexibility, with later methods building on top of FaceID, however, id-adapters trained for new diffusion models frequently rely on conventional FaceID approach and codebase (Team (2024)). Another group of methods such as DreamBooth Ruiz et al. (2023) and similar are also applicable to this problem, however, they are heavily limited due to need for finetuning for each new identity.

**Diffusion distillation** Diffusion distillation is an approach to accelerate trained diffusion models by training them to sample in few steps while still trying to model original $p_{data}(x)$ as close as possible (Salimans & Ho (2022), Song et al. (2023), Yin et al. (2024)). State of the art approaches such as LCM Luo et al. (2023) and Hyper Ren et al. (2024) remain common for new model releases (Ke et al. (2024), Chen et al. (2024)) ), but new distillation techniques are actively being developed. In practice, these distilled versions may differ in their inference qualities and sampling procedures, generally applicable in range of 1-8 sampling steps. Application of these distilled models to image conditioned generation and in particular id-preserving generation is at the heart of this work.

**Adaptation to new diffusion models** Cheap adaptation of pretrained modules for diffusion models to new checkpoints has been explored in Lin et al. (2024a) authors train an adapter module that

acts as a latent projection between the inner-layer connection of the original ControlNet and new diffusion model and they achieve fast generalization. In other work Xu et al. (2024) authors consider a case of efficient adaptation of ControlNet to new conditional domains. In the context of distilled diffusion models similar problems have also been explored with ControlNet: (Xiao et al. (2023), Parmar et al. (2024)), where in both works specific finetuning approaches are proposed either to match distillation objective or enforce cycle-consistency. Limitations of available solutions are either designing finetuning approach per checkpoint or tolerating baseline quality. We show that it is possible to universally boost quality of such adaptation without any additional training.

## 3  METHOD

### 3.1  BACKGROUND

We highlight three design choices that are commonly used in construction and training of ID-adapters. Firstly, Eq. 1 describes information flow introduced through additional cross-attention blocks. These additional blocks are called decoupled and introduce new keys and values $K'$ and $V'$ (for details see Ye et al. (2023)). Importantly, a single scalar value $\lambda$ controls the input of visual information from face, and we will address this in Section 3.3.

$$z_{new} = Attn(z; Q, K, V) + \lambda \cdot Attn(z; Q, K', V') \tag{1}$$

Secondly, a commonly known technique classifier-free guidance is adopted for two distinct conditions as described in Eq. 2. $x_t$ is omitted for clarity. It can be seen that scale parameter $w$ impacts both conditioning strength on $c_{text}$ and $c_{id}$, not allowing any flexibility given two distinct conditions.

$$\hat{\epsilon} = \epsilon(\varnothing, \varnothing) + w \cdot (\epsilon(c_{text}, c_{id}) - \epsilon(\varnothing, \varnothing)) \tag{2}$$

Lastly, we note that training id-preserving adapters trained via diffusion target (Ho et al. (2020)) are proximally trained to reconstruct identity in the image, i.e. maximize similarity between the person in $c_{id}$ and $\hat{x}_0$ given conditional information. However, during inference, this is not strictly the case. Given pretrained ID-adapter we identify two common generation purposes - *stylistic* and *realistic*. By "stylistic" we define generation that implies visual domain shift towards some priory known style, e.g. "pixel art" implies that generated image is expected to follow pixel-like visual appearance; by "realistic" setup we define generation that is not biased to any specific style or biased explicitly towards "realism" as specified in prompt. These cases correspond to different goals - in "stylistic" the user is less interested in facial features similarity and more in style following, while in realism situation is opposite, examples given in Figure 2.

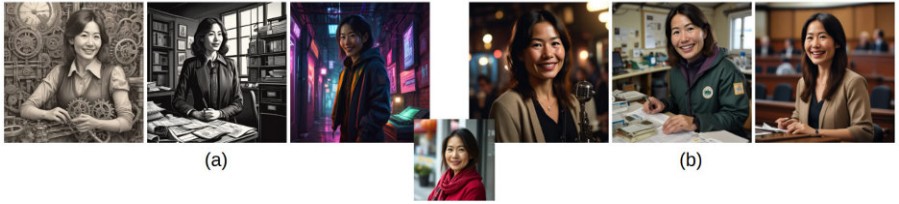

Figure 2: Different cases of user intention during ID-preserving generation: (a) - stylistic, (b) - realistic

In the following section we built components of FastFace on top of described design choices, and then unify them in one framework.

### 3.2  DECOUPLED CLASSIFIER FREE GUIDANCE

**CFG Decomposition**  Classifier-free guidance decomposition was introduced in Brooks et al. (2023) in the following form: $\hat{\epsilon} = \epsilon(\varnothing, \varnothing) + \alpha \cdot (\epsilon(c_{img}, \varnothing) - \epsilon(\varnothing, \varnothing)) + \beta \cdot (\epsilon(c_{text}, c_{img}) - \epsilon(c_{img}, \varnothing))$. In this equation we split guidance between $c_{text}$ and $c_{img}$ which enables to tune between preservation and editing in instruct editing task. However it remains understudied, since it has

not been widely adopted in later works (Labs et al. (2025), Zhang et al. (2025)) and has not applied before for generation with ID-adapters.

In identity-preserving generation we find same decoupling also works, while instead of $c_{img}$ we use $c_{id}$ with reference identity. We give additional possible derivation and ablations of this expression in in Appendix D. In this setup $\alpha$ corresponds to strength of id conditioning and $\beta$ corresponds to textual strength conditioning, however it is not yet applicable to distilled models.

**DCG - few-step stylistic tuning**  Simply substituting $c_{id}$ and changing $\alpha$ and $\beta$ will result in degradation and artifacts for distilled models not suited for guidance, see Appendix D for examples. Therefore we introduce two contributions to make it work with distilled models. Firstly we ablate scheduling regimes of conventional classifier-free guidance for our setup. Contrary to findings in previous works (Wang et al. (2024b), Starodubcev et al. (2024)) we find that scheduling should be applied only to intermediate steps, see Appendix D, and use it to bias model towards prompt following.

Secondly, to further enhance visual quality we apply rescaling to decoupled terms to balance norms of output and predicted noises, and , which is inspired by rescaling trick introduced in Lin et al. (2024b). Overall algorithm of DCG is given in equations below - in second expression $\sigma_i$ and $\sigma_{ti}$ correspond to standard deviation of corresponding conditional predictions $\epsilon(c_{id}, \varnothing)$ and $\epsilon(c_{text}, c_{id})$, and this deviations are averaged, last equation introduces interpolation trade-off between stability and quality, scaling hyper-parameter $\phi$ in practice is fixed.

$$\hat{\epsilon} = \epsilon(\varnothing, \varnothing) + \alpha(t) \cdot \underbrace{(\epsilon(c_{id}, \varnothing) - \epsilon(\varnothing, \varnothing))}_{\text{id guidance}} + \beta(t) \cdot \underbrace{(\epsilon(c_{text}, c_{id}) - \epsilon(c_{id}, \varnothing))}_{\text{text guidance}} \quad (3)$$

$$\epsilon_{\text{rescaled}} = \frac{\sigma_i + \sigma_{ti}}{2\hat{\sigma}} \hat{\epsilon}, \quad (4)$$

$$\epsilon_{\textbf{DCG}} = \phi \cdot \hat{\epsilon}_{\text{rescaled}} + (1 - \phi)\epsilon_{dcg} \quad (5)$$

After applying proposed changes, we can choose coefficients $\alpha$ and $\beta$ in wider range as demonstrated in Figure 3a. We choose $\alpha(t) = [1.0, 1.5, 1.5, 1.0]$ and $\beta(t) = [1.0, 3.0, 3.0, 1.0]$ and find them applicable to all studied checkpoints with "style" prompts, as it enhances coherence with described style at low-level details additional ablation with stylistic part of DiverseBench is given in Table 1.

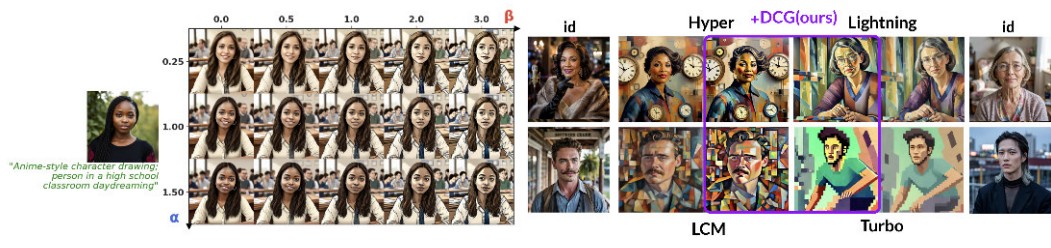

(a) Ablation grid of DCG with proposed fixes, only intermediate step coefficients are altered

(b) Visual result of applying DCG to stylistic generation with various models

Figure 3: DCG visualizations; (a) - ablation of coefficient demonstrate visual trade-off between tuning $\alpha$ and $\beta$, (b) - visual result of tuned DCG added to inference across different distilled checkpoints

| guidance setup | CLIP (↑) | IR (↑) | ID (↑) |
|---|---|---|---|
| const(baseline) | 0.268 | 0.901 | 0.258 |
| schedule | 0.276 | 1.278 | **0.320** |
| schedule + rescale | **0.277** | **1.289** | 0.318 |

Table 1: DCG components ablation with stylistic generation

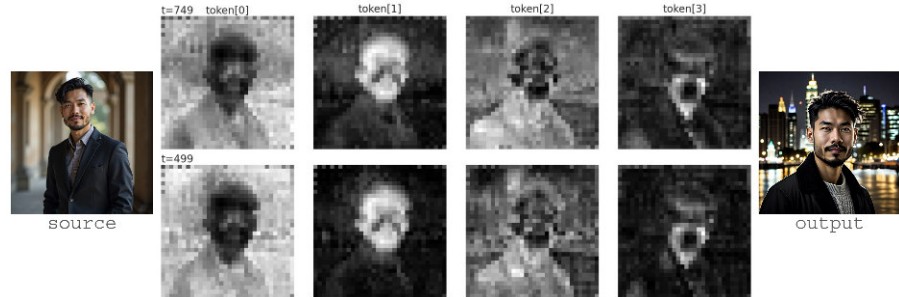

Figure 4: Visualization of attention maps timesteps 749 and 499 in decoupled block of SDXL in relation to generation output

### 3.3 ATTENTION MANIPULATION

**Motivation** Attention maps in diffusion models are known to contain a lot of semantic and spatial information, which has been applied in numerous works of image editing (Hertz et al. (2022), Cao et al. (2023), Epstein et al. (2023), Titov et al. (2024)). Nuance of ID-adapters is that they train new cross-attention blocks within UNet to condition on visual information from $c_{id}$. We inspect these new blocks and visualize attention maps in Figure 4 - it can bee seen that they share a lot of information with facial features and position in generated images, while also containing a lot of noisy signal about surrounding context, which can't be removed by changing `ip_adapter_scale` (see Eq. 1). Therefore we opt to work with attention maps directly.

**Basic formulation** We begin with formulation of general Attention Manipulation (AM) algorithm in Equation 6. Main challenge is to construct such $f(\cdot) : A \mapsto \tilde{A}$, where $A$ in attention map in decoupled blocks, that $\tilde{A}$ would allow achieve properties of 1) increasing face similarity/fidelity without significantly damaging prompt following 2) steering id-preserving generation towards more stable results, which we achieve by *focusing attention on face regions*.

$$softmax(\frac{Q(K')^T}{\sqrt{d}}) \longrightarrow f(A) \longrightarrow \tilde{A}V' \longrightarrow z \tag{6}$$

**Scale-power transform** First transformation is designed via simple composition of scale and power transform applied to attention maps. Detailed ablation of this operations is given in Appendix E, intuitevely power transformation applied to values less then 1 shifts everything closer to 0, while scaling linearly enhances attention mainly in meaningful tail of distribution with face region.

$$f_{sp} := (\text{scale} \circ \text{power})(A) = s \cdot A^p \tag{7}$$

**Steering scheduled-softmask transform** Second transformation is designed in more tricky way to steer generation towards more stable, portrait-like images on average. This purpose is motivated by presence of "failure" cases, where for some reason id-preserving generation deviates towards unrealistic imagery or fails to preserve features in meaningful way, therefore requiring more global transformation, examples are given in Appendix E. It is constructed of following components 1) firstly Equation 9 performs an adaptive distribution shift of values less then $Q_p(A)$ towards 0 and others towards 1, strength of shift is defined by parameter $d$ 2) $d$ is scheduled to large value at first step to influence global structure of the image 3) smooth alignment with original attention statistics inspired by AdaIN Huang & Belongie (2017) is applied - normalizing transformed attention maps, modulate them using $\mu_A$ and $\sigma_A$ of original maps and interpolate between modulated and transformed versions, same operation is also applied to output of attention block. Complete definition of $f_{ss}()$ is given in Equations below.

$$F(A, p) := \text{norm}(A) - Q_p(A) \qquad (8)$$

$$\text{SM}(A, d, p) := s \cdot \sigma(\text{norm}(\sigma(-dF(A, p)))) \qquad (9)$$

$$f_{ss}(A) := ws \cdot \text{SM}(A, d, p) + (1 - w)\text{AdaIN}(A, s \cdot \text{SM}(A, d, p)) \qquad (10)$$

In Equation 9 $norm(\cdot)$ denotes normalization and $Q_p(\cdot)$ is a p-th quantile function. In Figure 5 we visualize effect transforms have on attention values. We validate that proposed transforms achieve desired effect of enhancing identity similarity independently of face size in Figure 6.

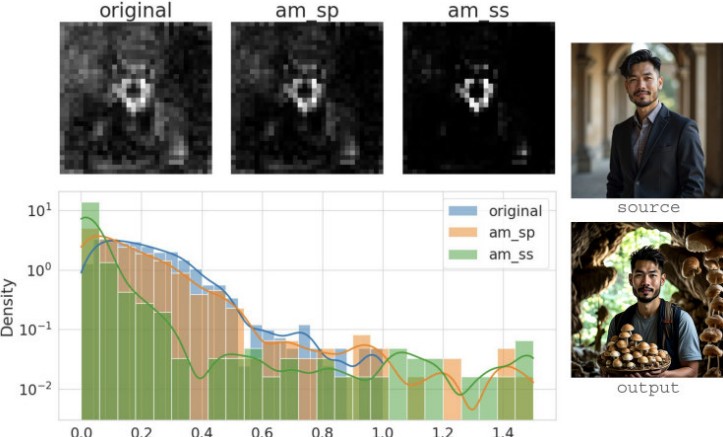

Figure 5: Visualizations of $f_{sp}$ and $f_{sm}$ transforms. At the top - visual result of transformation on the level of attention maps at certain block/step/token, bottom - distribution shift of attention values

We as well provide detailed ablations and visual results for AM setups, as well as sensitivity analysis in Appendix E and G.

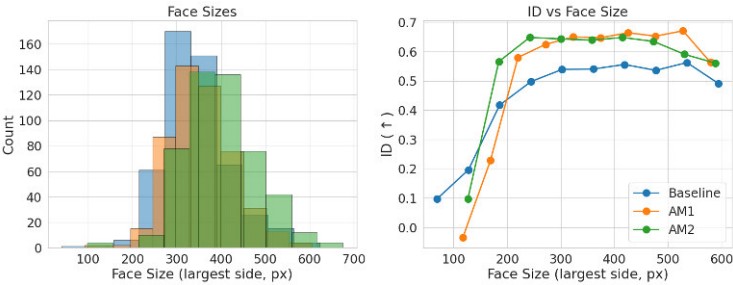

Figure 6: left - histogram of face sizes in baseline and different AM configurations, right - distribution of face sizes values in different setups (AM1 - scale-power, AM2 - scheduled-softmask)

### 3.4 FULL FRAMEWORK AND EVALUATION

Together, presented mechanisms formulate joint framework of FastFace - through use of DCG and AM, which can be applied together or independently to any few-step sampling models, and are visualized in Figure 7. In further sections we will demonstrate that these mechanisms work well together in general setting of id-preserving generation, as well as their respective setups of stylistic/realistic generations.

**Open evaluation** In recent works (Ye et al. (2023), Wang et al. (2024a), Guo et al. (2024)) authors rarely provide clarity about data which was used for evaluation, not allowing to fairly compare one method quality to another and understand their strength and weaknesses with respect to realistic or stylized generation with detailed prompts. When evaluating these methods we find that in a lot

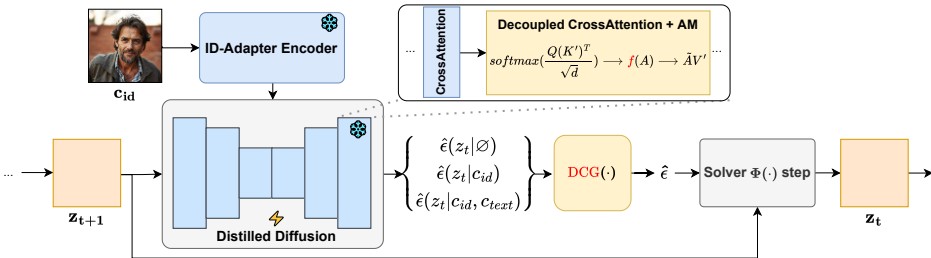

Figure 7: FastFace joint pipeline with proposed mechanisms - decoupled classifier free guidance, expanding on outputs of UNet, and attention manipulation as transform in decoupled blocks

of cases they fail completely in practical setting, see Appendix C. In this work stick to transparent evaluation in terms of identity images and prompts used.

**Dataset details** We collect a synthetic dataset DiverseBench consisting of 54 high-resolution identity images from several models, ensuring diversity and filtering by mean similarity threshold within identity groups. Prompts are constructed for two settings - 80 for realistic and 40 for stylistic. During evaluation, fixed amount of random pairs without replacement is sampled, covering a wide range of identity and prompts combinations. Additionally we use a subset from OmniContext dataset, which corresponds to single character generation with detailed prompts. Detailed description of collection and processing of DiverseBench is given in Appendix B.

## 4 EXPERIMENTS

In following experiments we try to answer two questions - *"does FastFace generalize across models"* and *"does FastFace generalize across ID-adapters"* without specific tuning of parameters per model/adapter. We do this by comparing against multiple known baseline which are attached to same checkpoints, and support presented metrics with multiple visual examples.

### 4.1 METRICS

**Common metrics** Metrics applied in both setups are face-similarity (ID), estimated as cosine distance between embeddings extracted by `buffalo-l` backbone from faces in source and generated images, and CLIP score (CLIP) between generated images and prompt computed with `clip/l-14` to estimate prompt alignment. We use LAION-Aesthetic (AE) reward model, which was trained on LAION subset, to estimate image quality/fidelity of image in both general evaluation and with realistic subset LAION (2022). Additionally we use ImageReward (IR) reward model to measure quality of stylistic images in stylistic ablations - as it was trained on synthetic data and is biased towards colorfulness and details and is more suited for that setting Xu et al. (2023). We account for `face_fail_cnt` (FFC) - an integer metric which value represents amount of cases where no face was detected in generated image - measure of how unstable certain method is.

### 4.2 RESULTS

Below in Table 2 we present evaluation of FastFace framework applied with FaceID-Plus-v2, as well as other common ID-adapters applied with different checkpoints. A more technical ablation over this and other checkpoints can be found in Appendix H. FastFace allows to achieve superior identity preservation, while achieving CLIP and AE metrics comparable with less constrained methods like PuLID and Photomaker and not requiring additional compute like RectifID.

In Figure 8 we present result of applying FastFace framework for both realistic and stylistic generation results. Beyond main metrics given in, FastFace is able to introduce stability into resulting face generation. We additionally analyze sensitivity of FastFace hyperparameters in Appendix H and show that it is robust across wide range of values.

Table 2: Metric comparison of baseline setup against FastFace setups - $FF_{AM1}$ denotes application of DCG with scale-power transform, $FF_{AM2}$ - DCG with scheduled-softmask transform

| Model | Method | DiverseFaces | | | | OmniContext | | | | |
|---|---|---|---|---|---|---|---|---|---|---|
| | | ID (↑) | CLIP (↑) | AE (↑) | FFC (↓) | ID (↑) | CLIP (↑) | AE (↑) | FFC (↓) | time(sec) |
| SDXL-Hyper | FaceID | 0.536 | 0.267 | 5.661 | 1 | 0.643 | 0.268 | 5.422 | 1 | 2.32 ±0.03 |
| | FaceID-Portrait | 0.354 | 0.259 | 5.736 | 25 | 0.581 | 0.251 | 5.328 | 0 | 2.13 ±0.01 |
| | FaceID-Plus-v2 | 0.588 | 0.259 | 5.889 | 0 | 0.646 | 0.269 | 5.665 | 0 | 2.41 ±0.04 |
| | PuLID | 0.238 | 0.262 | 5.902 | 72 | 0.335 | 0.277 | 5.828 | 0 | 2.56 ±0.38 |
| | Photomaker | 0.127 | 0.267 | 5.691 | 80 | 0.155 | 0.281 | 5.658 | 0 | 1.50 ±1.38 |
| | RectifID* | 0.369 | 0.263 | 5.322 | 6 | 0.283 | 0.288 | 5.309 | 2 | 16.87 ±4.73 |
| | **FastFace-AM1 (ours)** | **0.623** | 0.260 | 5.854 | 0 | **0.683** | 0.264 | 5.591 | 0 | 2.43 ±0.06 |
| | **FastFace-AM2 (ours)** | 0.590 | 0.242 | 5.882 | 1 | 0.627 | 0.260 | 5.704 | 0 | 2.42 ±0.05 |
| SDXL-Lightning | FaceID | 0.474 | 0.265 | 5.545 | 1 | 0.593 | 0.264 | 5.341 | 0 | |
| | FaceID-Portrait | 0.341 | 0.256 | 5.661 | 18 | 0.515 | 0.256 | 5.366 | 0 | |
| | FaceID-Plus-v2 | 0.517 | 0.259 | 5.715 | 0 | 0.590 | 0.271 | 5.511 | 0 | |
| | PuLID | 0.218 | 0.257 | 5.752 | 154 | 0.295 | 0.273 | 5.609 | 1 | |
| | Photomaker | 0.093 | 0.267 | 5.649 | 86 | 0.141 | 0.282 | 5.580 | 0 | |
| | **FastFace-AM1 (ours)** | **0.568** | 0.261 | 5.747 | 0 | **0.626** | 0.267 | 5.568 | 0 | |
| | **FastFace-AM2 (ours)** | 0.553 | 0.236 | 5.699 | 2 | 0.589 | 0.258 | 5.477 | 0 | |

\* evaluated as in original paper with 'perflow-sd15-dreamshaper'

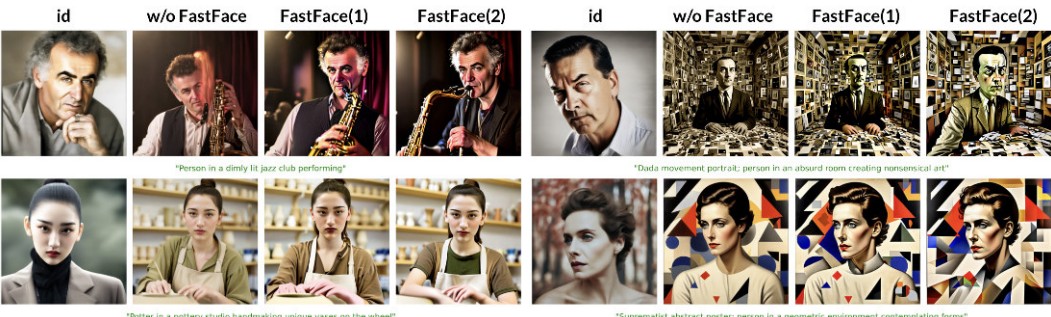

Figure 8: Demonstration of application of framework to real based identities from evaluation set

### 4.3 BEYOND FACEID-PLUS-V2

To further study generalization of proposed framework, we additionally evaluate it with standard FaceID and also try applying to one of the recent ID-adapters - PuLID. In Table 3 we numerically account for contribution of proposed methods towards better trade offs, and visual examples are given in Fig. 9. Note that PuLID generally finds current evaluation prompts challenging to follow while preserving any ID, but proposed framework still allows to boost methods quality.

## 5 CONCLUSION

This work presents lightweight and easy-to-implement FastFace framework, which solves problem of adaptation of pretrained id-preserving generation adapter to distilled diffusion model without additional retraining. Included methods are developed for different cases of id-preserving generation - "stylistic", to better match style described in prompt, and "realistic", to enhance identity similarity or fidelity of the image. Presented contributions are evaluated in general, as well in specific scenarios on constructed evaluation dataset for id-preserving generation, showing generally better trade-offs in terms of identity preservation, prompt following and image quality.

## 6 LIMITATIONS

Although proposed methods show promising results, scope of current work is limited to training-free methods, which are ultimately bottle-necked by distilled diffusion model checkpoint, and generally shows less impressive results in extreme cases such single-step sampling regime. It is a future work matter to address these limitation and adapt id-preserving generation to single-step models.

| Model | ID (↑) | CLIP (↑) | AE (↑) |
|---|---|---|---|
| **Hyper + FaceID** | | | |
| base | 0.580 | 0.251 | 6.047 |
| $FF_{AM1}$ | **0.595** | **0.258** | **6.125** |
| **Lightning + FaceID** | | | |
| base | 0.508 | 0.246 | 6.002 |
| $FF_{AM1}$ | **0.535** | **0.258** | **6.102** |
| **Hyper + PuLID** | | | |
| base | 0.179 | 0.262 | 6.207 |
| $FF_{AM1}$ | **0.227** | 0.261 | **6.261** |
| **Lightning + PuLID** | | | |
| base | 0.172 | **0.258** | 6.096 |
| $FF_{AM1}$ | **0.228** | 0.254 | **6.254** |

Table 3: Evaluation results for other id-adapters

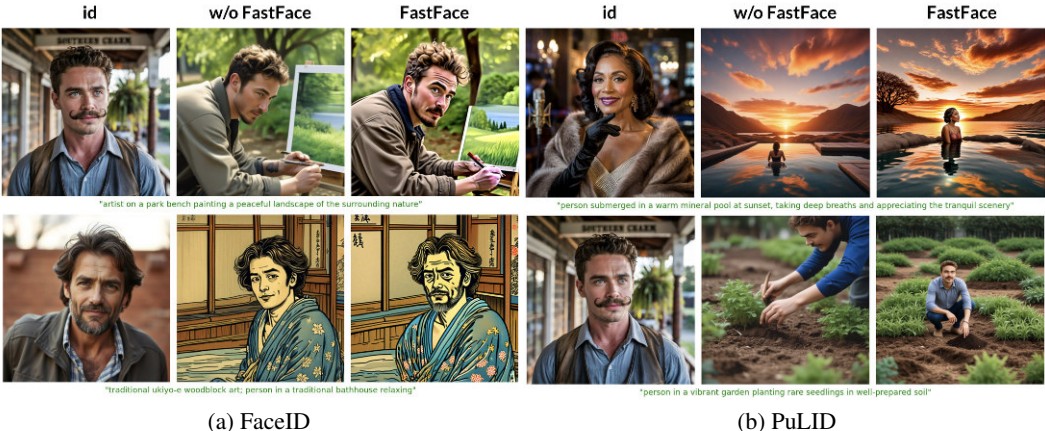

(a) FaceID                 (b) PuLID

Figure 9: Application of FastFace framework to other ID-adapters

## 7 REPRODUCIBILITY

This work ensures reproducibility through three primary measures: (1) detailed algorithmic descriptions within the manuscript, (2) release of evaluation datasets, and (3) the use of fixed random seeds to ensure deterministic experimental outcomes.

## 8 ETHICAL STATEMENT

This work presents methods for generating and manipulating human faces. All facial images used for evaluation in our benchmark, DiverseFaces, are synthetically generated. However, we acknowledge that the ability to realistically modify human likenesses carries inherent risks, including the potential for misuse to create misleading or harmful content. To mitigate this, we have chosen to use only high-quality and diverse data for our benchmark, avoiding the use of real individuals' likenesses without explicit consent. We strongly advocate for the responsible development and use of such technologies, including the implementation of robust safeguards, provenance tracking, and public education to prevent misuse.

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

## A APPENDIX

## B DETAILS OF EVALUATION DATASET

We develop an evaluation dataset consisting of 54 high quality identity images and 120 prompts, which are used as input conditions for generation and further evaluation. Identity images are synthetic images from models such as Flux and Ideogram 3.0 (Labs (2024)), representing different age groups (young, middle age and old), genders and ethnicities, examples are presented in Figure 11. Part of images was also synthesized using id-preserving methods with from real identities, thus avoiding bias towards only synthetic facial features. Additionally, to ensure variance within groups of identities of same gender and age, further cleaning was done by thresholding and replacing identity images with largest mean face similarity to others, i.e. if $\frac{1}{n-1}\Sigma_{j,j\neq i}sim(c_i,c_j) > 0.3$ for $c_i$ within group, it was discarded. Prompt description were also synthetically generated using Chat-GPT version of November 2024, generally following structure of `style` + `';'` + `'Person'` + `location` + `action`, and then additionally cleared and enriched. Prompts are categorized into two groups - 80 "realistic" prompts and 40 "style" prompts with certain style. Product of id images and prompts from category is considered as evaluation set, resulting in two sets - stylistic with 2160 and realistic with 4320 examples. Schematic depiction of the data collection is visualized in Figure 10.

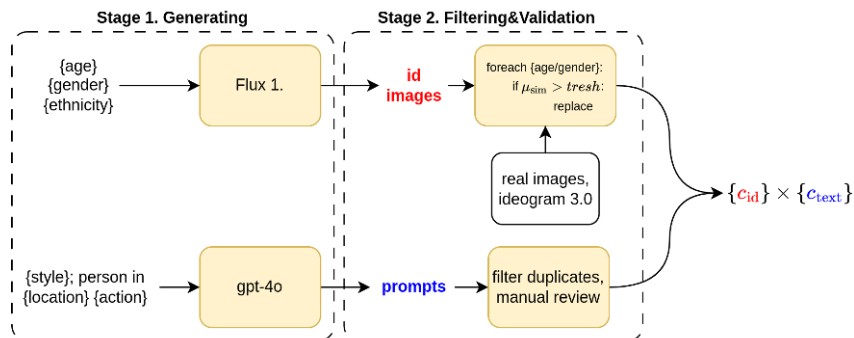

Figure 10: Evaluation dataset preparation pipeline

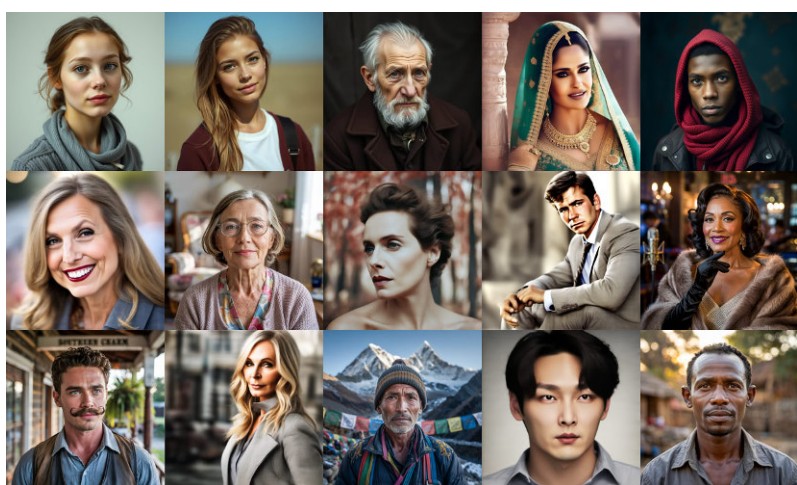

Figure 11: Evaluation dataset identity samples

## C CERTAIN ID-METHODS INFERENCE FAILURE EXAMPLES

Below we provide examples of recent id-preserving generation methods that we found to have limitations in terms of application with our evaluation set.

**PuLID** In Figure 12 we provide example common failure for PuLID method. From our experiments we find that it is not applicable with prompts that have description of context like location and action, which our evaluation set prompts have. We hypothesize that this effect is rooted in aligned training of PuLID, where inner representations of UNet are regularized to match generation without $c_{id}$ condition - in our experiments we found that in baseline setup FFC metric accounts around for 50% of sampled images failing (meaning around half of images doesn't have any identity detected).

**InstantID** This method is example of opposite behavior - it's pipeline includes ControlNet-like module that is conditioned on face key-points, which are extracted from source image by standard CV packages (e.g. `insightface` Insight (2023)). However, when tested against multiple different prompts, we observe in Fig. 13 that despite showing state of the art in terms of face preservation, outperforming any other method, it lacks prompt following and variability, not being able to properly follow details regarding background and person body position (additionally it has large bias towards watermark generation with 1:1 resolutions).

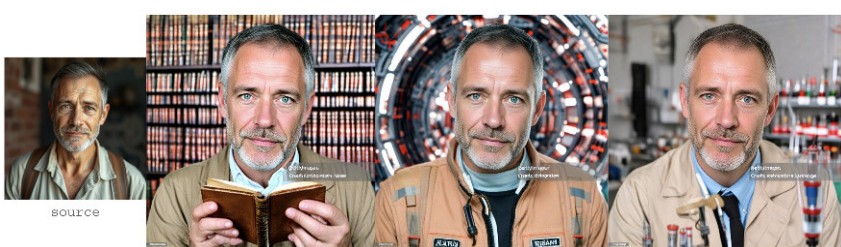

Figure 12: Demonstration of common case of failure for PuLID method - method lacks bias to human-centric generation to perform identity preservation, especially with small faces.

Figure 13: Demonstration of common case of failure for InstantID method - generated images are highly constrained and often omit details in the prompt, prompts used for generation: *"Person in an ancient library reading"*, *"Person in a futuristic space station repairing equipment"*, *"Person in a high-tech laboratory conducting experiments"*

## D DCG VARIATIONS AND DERIVATIONS

**Preliminary** To simplify derivation process let's recall that reverse diffusion process is formulated in terms of score function $\nabla_{x_t} \log p(x_t|y)$ Song et al. (2020), where $x_t$ is noised latent and $y$ is conditional information, in text2image models being prompt. Then classifier guidance can be derived as below, where in Eq. 13 $w$ is added as a hyper-parameter to control conditioning strength.

$$\nabla_{x_t} \log p(x_t|y) = \nabla_{x_t} \log\left(\frac{p(y|x_t)p(x_t)}{p(y)}\right) \tag{11}$$

$$= \nabla_{x_t} \log p(y|x_t) + \nabla_{x_t} \log p(x_t) - \nabla_{x_t} \log p(y) \tag{12}$$

$$\Rightarrow \nabla_{x_t} \log p(x_t) + w \cdot \nabla_{x_t} \log p(y|x_t) \tag{13}$$

Then to arrive to classifier-free guidance (which removes need for learning classifier $f(y|x_t)$ for estimation of $\nabla_{x_t} \log p(y|x_t)$), we rearrange terms in 13 and arrive to following:

$$\nabla_{x_t} \log p(x_t|y) = \nabla_{x_t} \log p(x_t) + w \cdot (\nabla_{x_t} \log(x_t|y) - \nabla_{x_t} \log(x_t)) \tag{14}$$

**DCG variants** Now let's derive possible decoupled classifier-free variants for two conditions, specifically when $y = [c_{text}, c_{id}]$. We note that $\nabla \log p(x_t|c_{text}, c_{id}) - \nabla \log p(x_t)$ from classifier-free guidance corresponds to estimation of $\nabla_{x_t} \log p(c_{id}, c_{text}|x_t)$ score function, which can be expressed in following ways:

$$\nabla_{x_t} \log p(c_{id}, c_{text}|x_t) = \begin{cases} \nabla_{x_t} \log p(c_{id}|x_t, c_{text}) + \nabla_{x_t} \log p(c_{text}|x_t) \\ \nabla_{x_t} \log p(c_{text}|x_t, c_{id}) + \nabla_{x_t} \log p(c_{id}|x_t) \\ \nabla_{x_t} \log p(c_{id}|x_t) + \nabla_{x_t} \log p(c_{text}|x_t) \end{cases} \tag{15}$$

Last expression is possible if we assume that $p(c_{id}, c_{text}) = p(c_{id})p(c_{text})$, which generally is not true, but since in practice choice of prompts and identities for id-preserving generation are not dependent, it can be valid. Finally, reformulating back to noise prediction, we arrive to three possible DCG formulations, where $DCG_2$ is the one used in main sections of the paper:

$$DCG_1(\hat{\epsilon}) := \epsilon(\varnothing, \varnothing) + \alpha \cdot (\epsilon(c_{text}, \varnothing) - \epsilon(\varnothing, \varnothing) + \beta \cdot (\epsilon(c_{text}, c_{id}) - \epsilon(\epsilon(c_{text}, \varnothing)) \quad (16)$$

$$DCG_2(\hat{\epsilon}) := \epsilon(\varnothing, \varnothing) + \alpha \cdot (\epsilon(\varnothing, c_{id}) - \epsilon(\varnothing, \varnothing) + \beta \cdot (\epsilon(c_{text}, c_{id}) - \epsilon(\varnothing, c_{id}) \quad (17)$$

$$DCG_3(\hat{\epsilon}) := \epsilon(\varnothing, \varnothing) + \alpha \cdot (\epsilon(c_{text}, \varnothing) - \epsilon(\varnothing, \varnothing)) + \beta \cdot (\epsilon(\varnothing, c_{id}) - \epsilon(\varnothing, \varnothing)) \quad (18)$$

In practice we find that expression in Eq. 17 works best in terms of semantic changes in the image. While Eq. 18 performs similarly, version in Eq. 16 suffers from image quality degradation and doesn't introduce smooth trade off between between identity preservation and stylization, see Figure 14.

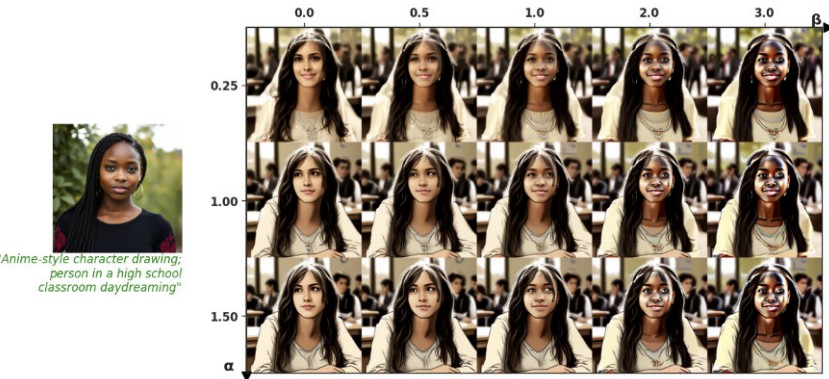

Figure 14: Ablation grid of DCG-1 variant, which can be observed to under perform compared to other decoupled options (same identity used as in Figure 3a)

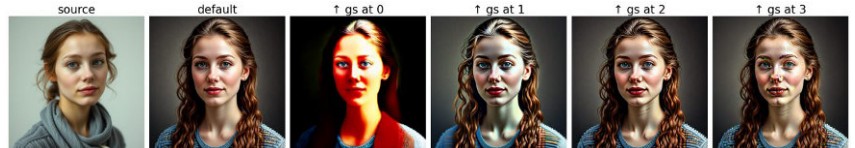

Figure 15: Scheduling effect on DCG, from right to left - baseline generation, single step alterations of $\alpha$ and $\beta$ coefficients to high value. In first steps image is completely corrupted, while last step introduces local visual artifacts

In Figure 15 we show that scaling at first and last steps results in significant artifacts, and below in Figure 16 we additionally provide visual examples of rescaling trick contribution in terms of local details in generated images. As it can bee seen, rescaling provides additional low-level enhancements of visual images in terms of details coherence.

## E    AM ANALYSIS AND DETAILS

**Scale-power ablation**    We provide visual ablation why scale-power transformation works in Figure 17. Scaling increases similarity, but alters image background, resulting in prompt following degradation. This is expected, as plugging scaling transform into Eq. 6 instead of $f()$ we can see that it is same as increasing $\lambda$. When raising attention values to some power, we achieve attention values shifting to 0, which decreases identity preservation, but increases prompt following, especially around face, since attention values in decoupled blocks stop interfering with attention from cross-attention blocks. Combination of transforms results in power transform basically canceling prompt following degradation of scale transform.

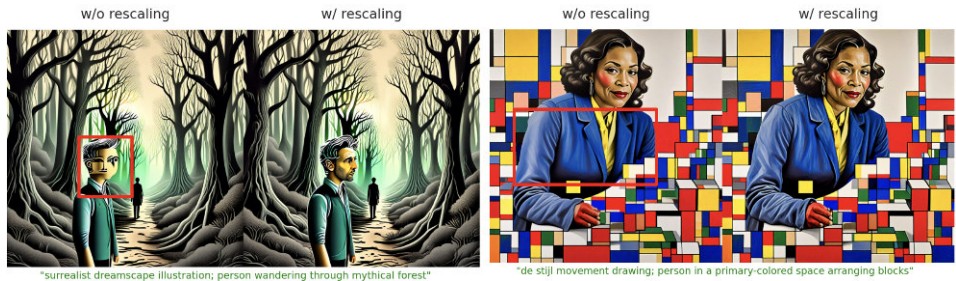

Figure 16: Effect of proposed rescaling on generated images with few-step models - areas with changes are highlighted

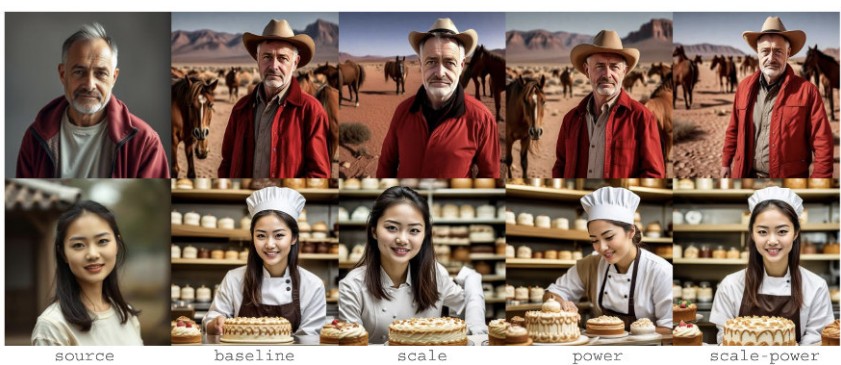

Figure 17: Visual ablation of scale-power transform components

**Failure cases demonstration** In Figure 18 we give examples of id-preserving failures with distilled diffusion model, where instead of expected outcome with human-centric generation method fails to preserve meaningfully align identity and surrounding context, which can result in identity morphing into background, being between multiple humans in image, unrealistic postures and etc. Such cases often can't be fixed by proposed scale-power transform, which serves as motivation for a more control-nature transform that changes structure of images.

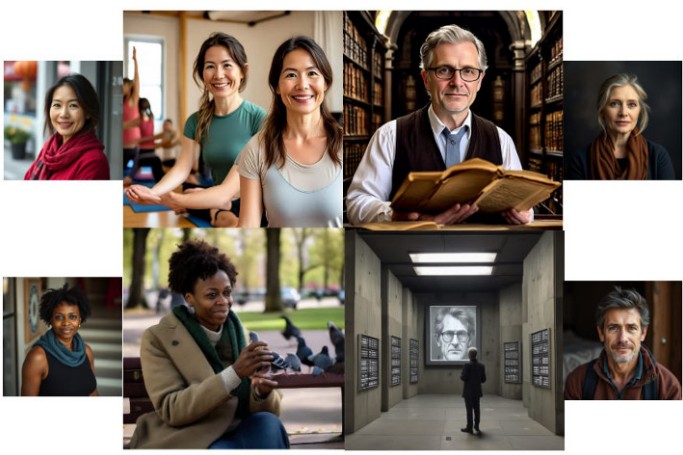

Figure 18: Generation examples with distilled model where generated image fails to successfully preserve identity in meaningful way

**Scheduled-softmask transform details** Beyond details provided in main sections, we also found that attention values for the first token in decoupled CrossAttention (see Fig.4) in FaceID-Plus-v2

are inverted - attention is focused on background across all blocks and timesteps, and it's values histogram has mode closer to 1 value. Therefore, when applying transformation to first token, we first invert it's values, and after transform invert back so that AM transformation has same expected effect across all tokens.

## F    RESULTS OF DCG IN STYLISTIC SETUP

In Figure 19 we present fronts for DCG in stylistic dataset for Hyper and Lightning. Parameters are specified in main section of the text are shares across all models and also joint application with AM.

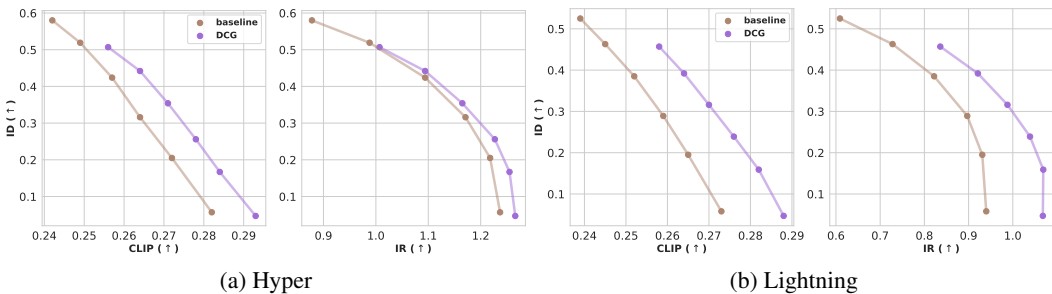

(a) Hyper                    (b) Lightning

Figure 19: Pareto fronts of Hyper and Lightning with DCG against baseline, stylistic setup, `lora_scale`= 1

In Table 4 we report metric comparisons for fixed `ip_adapter_scale`= 0.8 for all models. We can observe that DCG achieves expected degradation of face similarity, while increasing CLIP, IR and FCS.

Table 4: Ablation of DCG against baseline on stylistic data - DCG increases IR, CLIP score for general image and face area, while also bringing decrease in ID preservation

| **Model** | ID (↑) | CLIP (↑) | IR (↑) | FCS (↑) |
|---|---|---|---|---|
| Hyper | | | | |
| base | **0.519** | 0.249 | 0.988 | 0.180 |
| DCG | 0.442 | **0.264** | **1.094** | **0.184** |
| Lightning | | | | |
| base | **0.463** | 0.245 | 0.728 | 0.175 |
| DCG | 0.392 | **0.264** | **0.921** | **0.181** |
| LCM | | | | |
| base | **0.439** | 0.259 | 0.540 | 0.180 |
| DCG | 0.336 | **0.270** | **0.639** | **0.181** |
| Turbo | | | | |
| base | **0.310** | 0.252 | 0.888 | 0.165 |
| DCG | 0.254 | **0.277** | **1.007** | **0.175** |

## G    RESULTS OF AMS IN REALISTIC SETUP

Below we present results in terms of fronts computed on realistic subset and full table computed for fixed `ip_adapter_scale`= 0.8. `AM1` denotes scale-power transform and `AM2` denotes scheduled-softmask transform. In all setups (including joint application with DCG in following sections) all hyper-parameters are fixed across checkpoints and are following:

AM1 - target "up" and "down" unet parts, power strength $p = 1.3$, scale strength $s = 1.45$ in "down" part and $s = 1.55$ in "up" part.

AM2 - target "up" and "down" unet parts, scale strength $s = 1.55$ everywhere except first step; softmask quantile $p = 0.65$ softmask $d = 7.5$ at first step, $d = 5.$ at other steps; AdaIN blend coefficient $w = 0.7$.

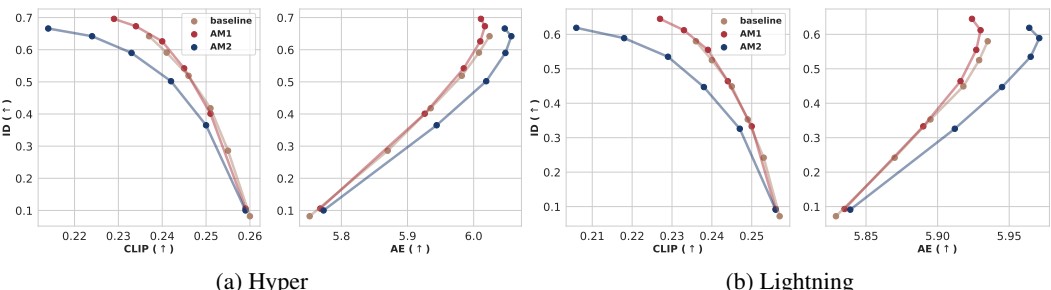

(a) Hyper

(b) Lightning

Figure 20: Pareto fronts of Hyper and Lightning with AM mechanisms against baseline, realistic setup

Table 5: Ablation of $AM$ transforms against baselines - Both $AM$ transformations increase identity preservation, aesthetic scores and stability, while slightly decreasing prompt following, $AM_2$ results in lower CLIP-score due to larger face bias

| Model | `lora_scale`$= 1.0$ | | | | `lora_scale`$= 0.5$ | | | |
|---|---|---|---|---|---|---|---|---|
| | ID $\uparrow$ | CLIP $\uparrow$ | AE $\uparrow$ | FFC $\downarrow$ | ID $\uparrow$ | CLIP $\uparrow$ | AE $\uparrow$ | FFC $\downarrow$ |
| Hyper | | | | | | | | |
| base | 0.591 | **0.241** | 6.008 | **0** | 0.408 | **0.255** | 6.229 | 19 |
| $AM1$ | **0.673** | 0.234 | 6.017 | 1 | **0.523** | 0.247 | 6.220 | 10 |
| $AM2$ | 0.642 | 0.224 | **6.057** | **0** | 0.517 | 0.239 | **6.265** | **3** |
| Lightning | | | | | | | | |
| base | 0.525 | **0.240** | 5.929 | 0 | 0.386 | **0.249** | 6.079 | 18 |
| $AM1$ | **0.612** | 0.233 | 5.930 | 0 | 0.494 | 0.241 | 6.088 | 12 |
| $AM2$ | 0.589 | 0.218 | **5.971** | 0 | **0.496** | 0.231 | **6.107** | **1** |
| LCM | | | | | | | | |
| base | 0.552 | **0.235** | 5.754 | 3 | 0.380 | **0.249** | 5.927 | 46 |
| $AM1$ | **0.610** | 0.227 | 5.783 | **1** | **0.477** | 0.240 | 5.942 | 34 |
| $AM2$ | 0.597 | 0.214 | **5.802** | **1** | 0.476 | 0.231 | **5.974** | **18** |
| Turbo | | | | | | | | |
| base | 0.349 | **0.243** | **5.650** | 94 | 0.189 | **0.250** | 5.764 | 116 |
| $AM1$ | **0.467** | 0.235 | 5.635 | 57 | **0.289** | 0.244 | 5.769 | 63 |
| $AM2$ | 0.443 | 0.230 | 5.647 | **62** | 0.283 | 0.240 | **5.784** | **51** |

In Figures 21 and 22 we demonstrate examples of applying just AM during inference. It can be seen that AM1 enhances identity similarity locally, without disrupting prompt following, while AM2 introduces larger faces and portrait like bias for generated outputs, which also results in lower prompt following.

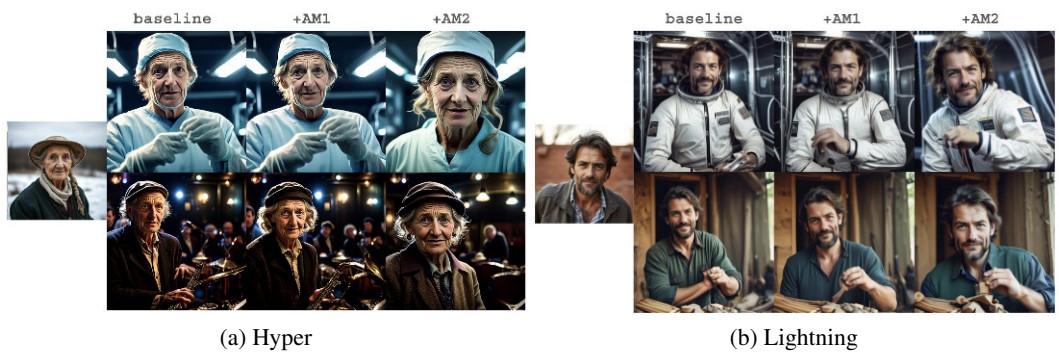

|     |     |
| --- | --- |
| (a) Hyper | (b) Lightning |

Figure 21: Application of AM compared to baselines

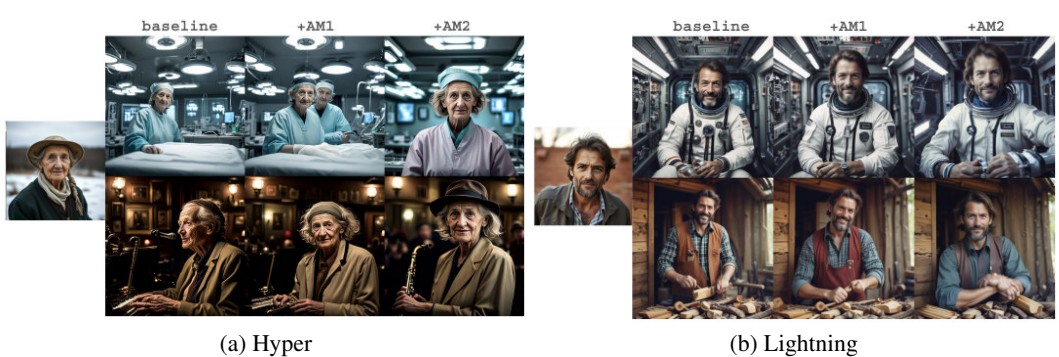

|     |     |
| --- | --- |
| (a) Hyper | (b) Lightning |

Figure 22: Application of AM compared to baselines, `lora_scale`=0.5

## H  ADDITIONAL FASTFACE RESULTS

### H.1  PARETO FRONTS

Below in Fig. 23 and 24 we provide Pareto fronts evaluated for FastFace framework on DiverseBench for varying `ip_adapter_scale` $\in \{0.1, 0.35, 0.5, 0.65, 0.8, 0.95\}$. These plots give additional information of scaling behaviors when trying to tune just ip-adapter scale. It can be seen that fronts introduced by FastFace achieve superior trade offs across different models.

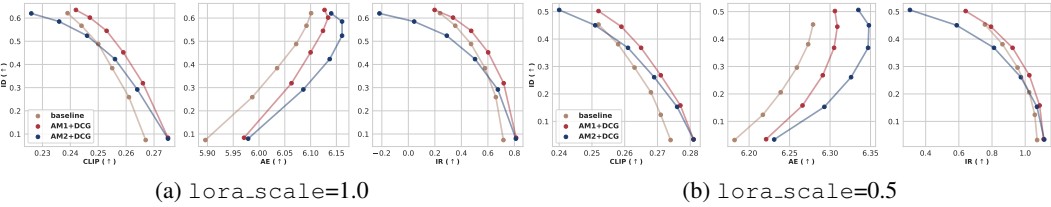

|     |     |
| --- | --- |
| (a) `lora_scale`=1.0 | (b) `lora_scale`=0.5 |

Figure 23: Pareto fronts built for Hyper model metrics with different scales of LoRA

### H.2  FASTFACE ABLATION

In Table 6 we report main metric evaluation for fixed value of $\lambda$ across all models with full and lower LoRA scale, common trick when applying ID-Adapters for more creative generation - as a result FastFace enhances identity similarity, image quality and stability without loss of prompt following.

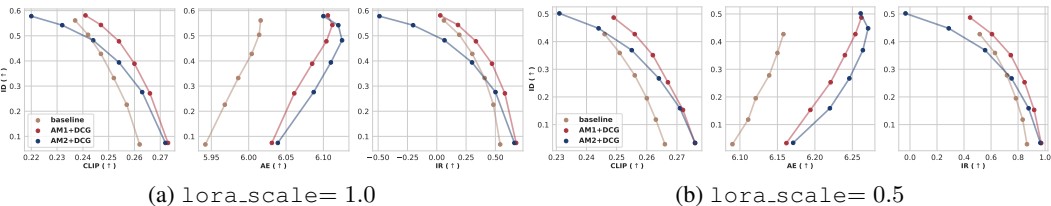

(a) `lora_scale= 1.0`    (b) `lora_scale= 0.5`

Figure 24: Lightning fronts for full data setup, different FastFace configurations and `lora_scales`

Table 6: Metric comparison of baseline setup against FastFace setups - $FF_{AM1}$ denotes application of DCG with scale-power transform, $FF_{AM2}$ - DCG with scheduled-softmask transform

| Model | `lora_scale= 1.0` | | | | `lora_scale= 0.5` | | | |
|---|---|---|---|---|---|---|---|---|
| | ID (↑) | CLIP (↑) | AE (↑) | FFC (↓) | ID (↑) | CLIP (↑) | AE (↑) | FFC (↓) |
| Hyper | | | | | | | | |
| base | 0.567 | 0.244 | 6.092 | 2 | 0.381 | 0.258 | 6.273 | 88 |
| $FF_{AM1}$ | **0.602** | **0.247** | 6.134 | 2 | 0.445 | **0.259** | 6.309 | 72 |
| $FF_{AM2}$ | 0.585 | 0.236 | **6.161** | **0** | **0.450** | 0.251 | **6.348** | **34** |
| Lightning | | | | | | | | |
| base | 0.504 | 0.242 | 6.014 | 4 | 0.359 | 0.251 | 6.150 | 89 |
| $FF_{AM1}$ | **0.543** | **0.247** | 6.112 | 2 | 0.427 | **0.256** | 6.254 | 66 |
| $FF_{AM2}$ | 0.542 | 0.232 | **6.120** | **0** | **0.448** | 0.244 | **6.271** | **33** |
| LCM | | | | | | | | |
| base | 0.515 | 0.243 | 5.770 | 53 | 0.344 | **0.258** | 5.911 | 288 |
| $FF_{AM1}$ | 0.525 | **0.244** | 5.796 | 37 | 0.383 | 0.258 | 5.968 | 202 |
| $FF_{AM2}$ | **0.533** | 0.229 | **5.807** | **20** | **0.406** | 0.246 | **5.979** | **136** |
| Turbo | | | | | | | | |
| base | 0.336 | 0.246 | 5.689 | 161 | 0.177 | 0.257 | 5.791 | **242** |
| $FF_{AM1}$ | **0.416** | **0.249** | 5.698 | 139 | 0.239 | **0.262** | 5.757 | 431 |
| $FF_{AM2}$ | 0.409 | 0.242 | **5.707** | **94** | **0.244** | 0.256 | **5.798** | 271 |

Additionally we provide sensitivity analysis of FastFace with respect to AM hyperparameters. It can be seen in the figure below that hyperparameters do not affect quality of the output in random way and can be chosen from wide range, offering optional tuning depending on the task and model.

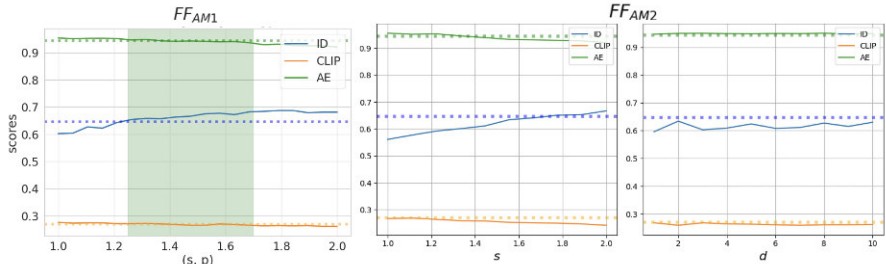

Figure 25: Sensitivity analysis w.r.t. to AM hyperparameters; left - $AM_1$, right - $AM_2$, AE metric is rescaled to match other metrics range between 0 and 1, dotted lines denote baseline quality

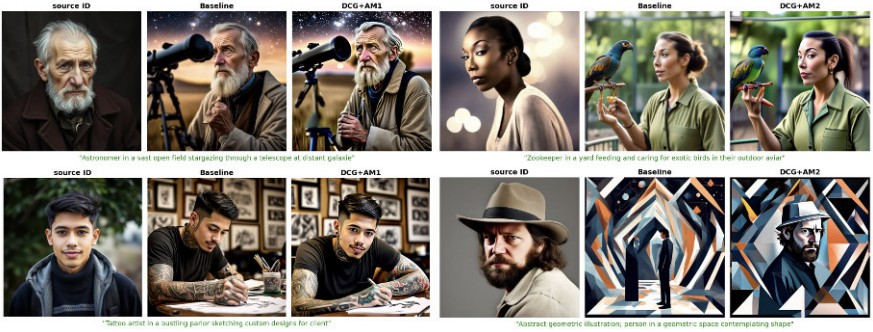

Figure 26: More inference examples of FastFace pipeline with IpAdapter-FaceID-v2 in different setups

