# OpenReview forum: "FastFace: Training-Free Identity Preservation Tuning in Distilled Diffusion via Guidance and Attention"
_ICLR.cc/2026/Conference — ICLR 2026 Conference Withdrawn Submission_

### Official Review · Reviewer_dV9u · 2025-10-30

**Soundness:** 3
**Presentation:** 1
**Contribution:** 2
**Rating:** 4
**Confidence:** 3

**Summary:**

This paper presents a set of training-free tricks to augment the distilled ID-consistent person generation model to achieve better ID preservation and preserve the instruction following ability.

For this objective, the paper presents two methods to augment adapter-based person generation models:

1. Decomposed Classifier-free guidance. The authors decompose the Classifier-free guidance into ID-preservation part and prompt-following part. By applying different weights and schedule for these two parts, existing person generation model can achieve better trade-off between ID conditions and text conditions.

2. Attention Manipulation. The authors find the token which corresponds to the face region in the generated images and design a set of denoising operations to make the attention map focusing more on face regions.

Experimental results on their curated benchmark and a subset of OmniContext dataset indicate their methods can effectively improve the identity preservation across two distilled ID-preserving person generation methods.

**Strengths:**

1. The idea of decomposing ID-preservation and prompt-following part in Classifier-free guidance sounds rational to me. Independent control of both parts can achieve better trade-off.
2. The proposed method achieves high efficiency after applying a set of test-time algorithms.
3. The proposed method can generalize to many adapter-based person generative models.

**Weaknesses:**

1. The writing of this paper requires improvement.
2. The evaluation looks weak. The proposed DiverseFaces benchmarks contains only synthetic data, which makes the evaluation biased. The single_character subset of OmniContext only has 50 samples for evaluation, making the results analysis less convincing.
3. The idea of attention manipulation is not novel. Previous works have explored the visualization of attention map like PuLID[1] and manually update attention map during inference like PersonaHOI[2]. How the proposed attention manipulation perform compared to existing methods is unclear. The motivation of designing such a complicated attention map denoising pipeline is also unclear for me.
4. The visualization has low quality, making it difficult to check quality of the generated images.

[1] PuLID: Pure and Lightning ID Customization via Contrastive Alignment. NeurIPS 2024.
[2] PersonaHOI: Effortlessly Improving Personalized Face with Human-Object Interaction Generation. CVPR 2025.

**Questions:**

How do you find the optimal number for so many new hyperparameters in different operations?

---

### Official Review · Reviewer_thNj · 2025-10-31

**Soundness:** 2
**Presentation:** 2
**Contribution:** 2
**Rating:** 2
**Confidence:** 5

**Summary:**

To address the incompatibility issue between existing ID adapters and distilled diffusion models—where either additional training is required or the performance is poor—this work proposes FastFace, a universal training-free framework. This framework achieves adaptation through two core mechanisms: 1) Decoupled Classifier-Free Guidance (DCG), which decomposes and optimizes guidance terms to support few-step stylistic generation (e.g., 4-step sampling); 2) Attention Manipulation (AM), which enhances attention on facial regions in decoupled blocks via scale-power transformation (AM1) and scheduled-softmask transformation (AM2) to improve identity similarity. Experiments have verified the effectiveness of this method.

**Strengths:**

1. This work addresses the incompatibility between existing ID adapters (e.g., FaceID, PuLID) and distilled diffusion models (e.g., SDXL-Hyper, Lightning) without requiring any additional training.
2. FastFace demonstrates robust compatibility with a wide range of distilled diffusion models and mainstream ID-preserving methods.
3. This paper is well-written and easy to follow.

**Weaknesses:**

1. Lack of validation on real human face datasets. All identity images used in the DiverseFaces benchmark—FastFace’s primary evaluation dataset—are synthetically generated rather than real human faces.
2. Limited compatibility with non-SDXL-based distilled models. FastFace’s experiments are exclusively conducted on SDXL-derived distilled models. There is no validation of its performance on distilled models from other base architectures.
3. There is no testing on extreme scenarios, such as low-quality reference images (blurred, occluded, or low-light), which can validate the robustness of the method.
4. No exploration of multi-reference image scenarios. FastFace’s entire design and evaluation focus on single-reference-image ID preservation. There is no experimental exploration of multi-reference setups (e.g., adapting to an identity using 2-5 reference images), which are common in practical personalized generation.

**Questions:**

1. Lack of validation on real human face datasets. All identity images used in the DiverseFaces benchmark—FastFace’s primary evaluation dataset—are synthetically generated rather than real human faces.
2. Limited compatibility with non-SDXL-based distilled models. FastFace’s experiments are exclusively conducted on SDXL-derived distilled models. There is no validation of its performance on distilled models from other base architectures.
3. There is no testing on extreme scenarios, such as low-quality reference images (blurred, occluded, or low-light), which can validate the robustness of the method.
4. No exploration of multi-reference image scenarios. FastFace’s entire design and evaluation focus on single-reference-image ID preservation. There is no experimental exploration of multi-reference setups (e.g., adapting to an identity using 2-5 reference images), which are common in practical personalized generation.

---

### Official Review · Reviewer_f74h · 2025-11-01

**Soundness:** 2
**Presentation:** 3
**Contribution:** 2
**Rating:** 4
**Confidence:** 3

**Summary:**

This paper proposes FastFace, a training-free framework for identity-preserving tuning in distilled diffusion models. The method consists of two components: the Decoupled Classifier-Free Guidance (DCG) module, designed for stylistic generation, and the Attention Manipulation (AM) module, designed for realistic generation. Both modules operate solely during inference and require no additional training. They can be directly applied to various distilled diffusion models and ID adapters, enabling fast, few-step identity-preserving generation.

**Strengths:**

1. Both mechanisms work at inference time and can be applied across multiple distilled diffusion models and ID adapters without retraining.
2. Maintaining identity consistency in distilled diffusion models is a practical and novel problem. The authors divide the task into stylistic and realistic goals and design DCG and AM modules respectively, which is clear and targeted.

**Weaknesses:**

1. The method proposed in this paper lacks theoretical rigor, and many of its design choices are derived purely from empirical experience rather than from complete theoretical derivation. For instance, in Appendix D, the authors acknowledge that their derivation relies on an independence assumption that “generally is not true,” which leaves the method without a solid theoretical foundation. Moreover, DCG is applied only at the intermediate steps, yet no theoretical explanation is provided for this choice; the authors merely observe through visualization that applying DCG at the first or last step leads to artifacts and therefore adopt this empirical rule. Although these design decisions may be effective in practice, they lack the necessary analytical justification and theoretical reasoning to support their validity.
2. The paper lacks a systematic ablation study of the DCG and AM modules. The authors clearly distinguish their respective roles: DCG is designed for stylistic generation and prompt-following, while AM focuses on realistic generation and identity preservation. However, in the experiments, both modules are directly combined and evaluated as a unified framework, making it difficult to quantify their individual contributions. Although the appendix provides partial ablation results for AM and DCG, it lacks comparative experiments conducted under identical conditions, for example, evaluating the complete performance metrics when using only DCG or only AM. Consequently, the current experimental setup is insufficient to reveal the specific contribution and mutual relationship of the two modules in the overall performance improvement.
3. The evaluation dataset used in this paper, DiverseBench, is overly idealized, as the experiments are conducted primarily on synthetic identities and synthetic prompts. All samples are generated by diffusion models and therefore lack the complexity found in real-world human faces, such as variations in lighting, pose, occlusion, and facial expression. Since all experiments are performed in such a controlled environment, the model’s generalization and robustness in real-world settings remain unverified. Consequently, the reported results mainly reflect performance under idealized conditions rather than stability in practical applications.
4. Although the authors describe the fundamental principles and respective roles of DCG and AM, they do not explain why combining the two modules leads to the best overall performance. According to the results presented in the appendix, using DCG alone increases the CLIP score while decreasing ID similarity, whereas using AM alone improves ID similarity but lowers the CLIP score. This suggests that the two modules respectively favor stylistic coherence and identity fidelity, and that using either one individually merely achieves a trade-off between different performance metrics. However, the paper does not provide a theoretical or mechanistic explanation of why their combination can simultaneously improve both metrics, nor does it discuss whether conflicts may arise between the two modules during inference. Therefore, the joint use of DCG and AM appears to be an empirical engineering design rather than a theoretically grounded method.
5. Equations (16) and (17) appear to be incorrect. Both expressions contain mismatched parentheses, and Equation (16) additionally includes an extra “ϵ”.

**Questions:**

1. Can the proposed method be applied to preserve non-facial human characteristics? AM enhances facial feature retention, but human characteristics are not limited to facial features—they also include factors such as body shape. Does AM have any effect on preserving these other features?
2. Can validation be conducted on real faces or public face datasets to confirm that the method can maintain stable ID similarity under complex, real-world conditions?
3. Please explain the interaction between DCG and AM. In some cases, when DCG and AM are used together, could their respective gains counteract each other? Are there situations where using only DCG or only AM yields the best performance?

---

### Official Review · Reviewer_dzJp · 2025-11-01

**Soundness:** 4
**Presentation:** 3
**Contribution:** 3
**Rating:** 6
**Confidence:** 3

**Summary:**

The paper presents a method for facial id preservation in distilled image generation using SDXL. This is a useful yet an underrepresented topic in research. The paper proposes well motivated though rather minimal insights as to how to implement this properly for the cases of realistic and stylistic generations. Most justifications are perfectly reasonable and justified. In terms of evaluation, the paper compares to rather old models - restricted to SDXL distillations, with very little qualitative results. This is a marginally good paper with some insights that might benefit the community, even though it is a bit dated with incremental contribution.

**Strengths:**

- Simple to implement, understand, and justify insights.
- Results seem to improve upon the baselines.

**Weaknesses:**

- Identity preservation is still far from perfect.
- Comparisons and implementation are rather dated.

**Questions:**

- How would this approach affect newer models, employing flow matching and hybrid attention? (such as Flux, SD3, QWEN etc.)
- Specifically, Qwen Image Edit supports multiple images, including identity, and has a small iteration count LoRA distillation, so that is an interesting comparison.
- Significantly many more qualitative examples should be added.
- I didn't understand the scale * power transform. Please explain in more detail.
- How would this approach compare against contemporary personalization method, such as anyStory, DreamO, or others?
- Why is the bold facing and underlining in the tables inconsistent? It seems a number is boldfaced or underline only if it is in your method's rows. That is misleading.

**Details Of Ethics Concerns:**

Every paper addressing personalization and specific human likeness is problematic in terms of ethics. This addressed shortly in the manuscript, and I guess that is sufficient.

---

### Note · Authors · 2025-11-14

I have read and agree with the venue's withdrawal policy on behalf of myself and my co-authors.